# Use of the Nursing Interventions Classification and Nurses’ Workloads: A Scoping Review

**DOI:** 10.3390/healthcare10061141

**Published:** 2022-06-19

**Authors:** Claudio-Alberto Rodríguez-Suárez, Martín Rodríguez-Álvaro, Alfonso-Miguel García-Hernández, Domingo-Ángel Fernández-Gutiérrez, Carlos-Enrique Martínez-Alberto, Pedro-Ruymán Brito-Brito

**Affiliations:** 1Insular Maternal and Child University Hospital Complex, Canary Health Service, 35016 Las Palmas de Gran Canaria, Spain; 2Santa Cruz de La Palma Primary Health Care Centre, Canary Health Service and Nursing Department, University of La Laguna, 38700 Santa Cruz de La Palma, Spain; 3Nursing Department, Faculty of Healthcare Science, University of La Laguna, 38200 Santa Cruz de Tenerife, Spain; almigar@ull.edu.es; 4Primary Care Management of Tenerife, Canary Health Service and Nursing Department, University of La Laguna, 38200 Santa Cruz de Tenerife, Spain; domingofernandez.gaptf@gmail.com (D.-Á.F.-G.); cmaalbp@gobiernodecanarias.org (C.-E.M.-A.); 5Training and Research in Care, Primary Care Management of Tenerife, Canary Health Service, 38200 Santa Cruz de Tenerife, Spain; ruymanbrito@gmail.com

**Keywords:** standardized nursing terminology, workload, review, nursing

## Abstract

Background: The Nursing Interventions Classification allows the systematic organisation of care treatments performed by nurses, and an estimation of the time taken to carry out the intervention is included in its characteristics. The aim of this study is to explore the evidence related to the use of the Nursing Interventions Classification in identifying and measure nurses’ workloads. Methods: A scoping review was conducted through a search of the databases Ovid Medline, PubMed, Web of Science, CINAHL, Scopus, LILACS and Cuiden. The DeCS/MeSH descriptors were: “Standardized Nursing terminology” and “Workload”. The search was limited to articles in Spanish, English and Portuguese. No limits were established regarding year of publication or type of study. Results: Few reports were identified (*n* = 8) and these had methodological designs that contributed low levels of evidence. Research was focused on identifying specific interventions, types of activities, the prevalence of interventions and the time required to perform them. Conclusions: The evidence found on determination of nurses’ workloads using the Nursing Interventions Classification was inconclusive. It is essential to increase the number of reports, as well as the settings and clinical context in which the Nursing Interventions Classification is used, with greater quality and methodological rigour.

## 1. Introduction

The Nursing Interventions Classification (NIC) allows the systematic organisation of care treatments carried out by nurses [1]. Since it first appeared in 1987, it has grown and been continuously developed as a result of the additions and reviews contributed by nurses. Among its features is an estimation of the time needed to perform the intervention and the minimum level of training the professional must have to carry it out safely and competently. The time needed to perform a NIC has been defined as the average time required to carry it out; this is an average rate that can be used to determine the remuneration rates derived from the nursing activity, long enough to carry out the intervention, although not so long that the economic costs are unreasonably high due to its remunerative effects [2]. Interventions were grouped into five categories: 15 min or less, 16–30 min, 32–45 min, 46–60 min and more than an hour. These estimates are based on the judgment of professionals who are familiar with the intervention and with the clinical specialty, and these may differ according to profession and settings. However, this offers a starting point from which to calculate the time and degree of training required, along with the cost of nursing care [2].

Cruz et al. [2] describe the concept of “workload” as the volume of nursing services provided by a care unit. This figure is obtained by measuring the care time dedicated to the nurses’ actions and multiplying them by the number of patients treated. The workloads described in the NIC do not correspond to simple activities, independent of the complexity of critical thought immersed in the nursing process. Individual and unit experience should be taken into account, along with other contextual factors that determine the outcomes of these workloads. Cordova et al. [3] observed a substantial reduction in the times reported by nurses to complete each of the interventions compared to those published in the NIC. This does not mean that the times are not valid; on the contrary, it reflects the highly specialised nature of the care administered in specific units. It is possible that the nurses in the units observed needed less time to complete most of the interventions as they were providing similar care to most patients. According to Cruz et al. [2], the activities carried out by the nurses could be “direct care” for the patient through immediate interaction involving physiological and psychosocial activities and including practical interventions and counselling support. On the other hand, “indirect care” activities involve actions related to management of the unit or interdisciplinary cooperation for the patient’s benefit [1]. From this perspective, the studies analysed indicate that the nurses dedicate 22–38% of their time to direct care, while indirect care represents some 26–50% [2]. Regarding anticipation of interventions, these can be classified as “scheduled” during the work shift, and “unscheduled”, which correspond to those that are unforeseen or cannot be predicted during the working day [3]. Separately, there are “non-specific” activities in the nursing profession that do not correspond to NIC taxonomy concepts and can be classified as “associated activities”. Similarly, activities carried out by the nurses during their working day related to meeting their own physiological needs or others of a personal nature are classified as “personal activities”.

The use of standardised nursing terminology, such as that in the NIC, allows nurses’ work to be represented in a uniform manner in IT systems, which is the first step in developing a measurement of workload and, at the same time, facilitates research into the effectiveness of care [3]. However, to determine actual workloads, it is essential to develop specific models that contain information on professional staffing. The NIC terminology only provides the bases that can be used to obtain a valid measure of nurses’ workloads. At this moment, measurement of nurses’ workloads available in the scientific literature using the NIC has only been reported in specific hospital situations [4,5], such as paediatrics [6,7,8] and oncology [9,10] clinical settings.

In the systematic review carried out by Cruz et al. [2], which had the main aim of synthesizing the evidence related to the use of the NIC to identify nurses’ workloads, the scarcity of studies, together with the low quality of the methodological designs used, produced inconclusive results. These authors classified the areas of interest in the studies included in their work into four groups: identification of specific interventions, prevalence of interventions in a particular setting, distribution of interventions according to type, and estimated time required to perform the interventions. Measuring the nurses’ workload is necessary to understand the real needs of nurses in health care systems, the activities they perform and the time they spend caring for patients in different situations. In this sense, the NIC is a chief resource for measuring workloads and supporting clinical management of nursing staff in hospitals and health care settings [2]. However, it has not been widely used in clinical practice [3]. Currently, there is little evidence in the literature on the use of average time ratios established by the NIC to measure workloads [2]. Owing to this scant evidence, it is crucial to identify studies with distinct designs that provide information, with a complementary vision that adjusts to a methodological structure [11] and for which an exploratory or scoping review has been carried out, with the aim of understanding the current state of the science regarding the following research question: Are there studies on the measurement of workloads through the use of NIC that contribute new evidence?

The main research aim is to explore the evidence related to the use of the NIC to identify and measure nurses’ workloads.

## 2. Materials and Methods

Design: the selected methodology was a scoping review adjusted to the Preferred Reporting Items for Systematic Review and Meta-Analysis Extension for Scoping Reviews (PRISMA-ScrR) Statement [12]. This methodological framework was proposed by Arksey & O’Malley (2005) and implemented by the Joanna Briggs Institute (JBI); our study aims to find gaps and scope the knowledge of the evidence related to using the NIC to identify and measure nurses’ workloads for which this methodology is most appropriate [13,14]. The research protocol was registered at the Open Science Framework (https://osf.io/6gv4t/ (accessed on 1 February 2022)). Inclusion criteria: studies published up to the 31st of December, 2021 in Spanish, English and Portuguese that approached the measurement of nurses’ workloads with the NIC in the international context. No limit was established concerning the year of publication or type of methodology. Exclusion criteria: studies not using NIC terminology. Sources of information: searches were carried out during the month of January, 2022 of the Ovid Medline, PubMed, Web of Science (WOS), CINAHL, Scopus, LILACS and Cuiden health science databases. *Search strategies:* the DeCS/MeSH descriptors used were: “Standardized Nursing Terminology” and “Workload”. Study selection and classification: all references were exported to Mendeley Reference Manager Online. Once the search was complete, we proceeded to eliminate duplicates and start screening for title and abstract. Subsequently, the full texts of the selected studies were examined to assess their eligibility, taking inclusion and exclusion criteria into account. Definition of study variables: bibliometric variables on the affiliation of the included studies; methodological quality and degree of scientific evidence; and content information variables. Data extraction: the studies identified were evaluated by 6 reviewers (A.M.G.H., C.A.R.S., C.E.M.A., D.A.F.G., M.R.Á. and P.R.B.B.). To assess study quality, JBI critical reading tools were used. Each of the researchers reviewed all the studies included and carried out data extraction independently. Relevant information was then pooled. To organize the presentation of results, the criteria established by Cruz et al. [2] were followed in four categories: identification of specific interventions, distribution of interventions by types of activities, prevalence of interventions, and estimated time required to perform the interventions.

## 3. Results

Studies identified were *n* = 79 following elimination of duplicates (*n* = 27) and those that did not correspond to scientific documents (*n* = 1). The number of studies was limited to *n* = 51, of which *n* = 40 were excluded following the screening of title and abstract. Next, the full texts of all the studies meeting eligibility criteria were retrieved (*n* = 11); *n* = 3 were excluded as they did not meet inclusion criteria. The number of reports included in the scoping review was *n* = 8, as can be seen in the flow-chart in Figure 1.

The date of publication was between 2010 and 2021. Most research was carried out in Brazil, with one study in the United States and another in China; publications were retrieved from six different scientific journals. The methodology used mainly followed a descriptive, observational design, while one employed qualitative methodology with the Delphi technique. All this information, the clinical setting and the research aims are shown in Table 1.

### 3.1. Identification of Specific Interventions

Various strategies were employed to identify the nursing interventions. Some studies identified the interventions corresponding to their respective clinical scenarios without performing mapping and validation processes, while other studies established a systematic procedure for the identification of activities through validation and mapping with NIC terminology through expert consensus, as shown in Table 2. 

### 3.2. Distribution of Interventions by Types of Activities 

There was interest in understanding the nature of the activities that generate nurses’ workloads. Some studies specified the following type of activities: scheduled, unscheduled, direct, indirect, associated and personal, as shown in Table 3.

### 3.3. Prevalence of Interventions

The prevalence offered proportions regarding the number of times each of the interventions described was carried out. Just two articles included in the scoping review provided data on the prevalence of interventions by nurses. 

Possari et al. [10] indicated the number and percentage of times each of the interventions was performed in the context of perioperative care in oncology surgery, distinguishing between direct (*n* = 380) and indirect care (*n* = 373). The most frequently recorded direct intervention was 6650 Vigilance (*n* = 76; 8.56%), and indirect was 7920 Documentation (*n* = 166; 18.69%). 

The number of interventions recorded by Somensi et al. [4] in the context of a hospital ward corresponded to the data obtained at two time points, through in-person and telematic observation. The direct care with the highest prevalence corresponded to 7400 Orientation in the health system, *n* = 230 (57.21%) in in-person observation and *n* = 150 (44.91%) in telematic observation. Among the indirect interventions with the highest prevalence, 7960 Exchange of health care information (*n* = 370; 24.68%) stood out in the case of in-person observation. Finally, in telematic observation, the prevalent intervention was Prescription registration (*n* = 384; 32.68%), which did not have a NIC code. The frequencies of all interventions carried out can be seen in Table 4.

### 3.4. Estimated Time to Carry out Interventions

The time that nurses dedicated to carrying out care interventions represents the final category analysed. Just two articles included in the scoping review provided data on the average time of interventions by nurses.

Cordova et al. [3] estimated the time required to perform each NIC, showing a minimum value = 8.45 min and a maximum value = 38.31 min (mean = 19.6; SD = 8.10). A total of *n* = 11 interventions were completed in accordance with the NIC, while *n* = 22 required more time and *n* = 12 less. Among the scheduled interventions (*n* = 25), *n* = 10 were completed with a similar average time to that indicated by the NIC, *n* = 5 required more time and *n* = 10 less. With regard to unscheduled interventions (*n* = 17), *n* = 1 was completed in accordance with NIC averages, *n* = 4 required more time and *n* = 12 less.

Somensi et al. [4] compared the average times taken by nurses to perform each of the care interventions under in-person observation (*n* = 1901) and telematic observation (*n* = 1509) without finding significant differences for direct (*p* = 0.427) or indirect (*p* = 0.486) in contrast to the averages established by the NIC. All average times can be seen in Table 5.

Finally, Trovó et al. [5] studied a patient sample (*n* = 200) transferred between units under intervention 7890 Transport: between facilities. The NIC establishes that the average time needed to carry out this intervention is 16–30 min. The mean time spent by nurses varied between 9.3 (SD = 3.5) and 12.2 (SD = 2.5) min, while transfers carried out by nursing assistants took between 7.1 (SD = 2.8) and 11.0 (SD = 2.2) min. Among the characteristics examined during the transfers of these patients, 46% of professionals omitted aspects regarding clinical information exchange and patients’ administrative documents, as well as neglecting transferred patients’ care needs in up to 39% of cases.

## 4. Discussion

The process for mapping activities in standardised language for interventions is used by 80% of studies in the literature on the basis of studies that use cross-mapping procedures through a focus group technique with specialist nurses, which consists of selecting a NIC intervention for each nursing activity, taking into account the similarity between the item and the definition of the intervention [2]. In this sense, a part of the nurses’ records are not clearly defined for mapping with normalised terminology, so some of the records cannot be evaluated according to the NIC time averages, which hampers the potential of the NIC to estimate actual workloads. The studies included have shown a high recording of direct and indirect activities and interventions. However, many of them need to be further standardised with the NIC terminology, as shown in the prevalence of interventions in Table 4 (interventions not identified with a NIC code). Currently, the high degree of computerisation of the Electronic Medical Clinical history in health services requires the application of mechanisms that facilitate the mapping and standardisation of records in IT systems. It is essential that the computerised system reaches its potential to fully capture the records [3] and that this information can be available and used to assess nursing activity.

Concerning to the average times required by nurses to perform the interventions, the studies by Cordova et al. [3] and Somensi et al. [4] reported the values and confidence intervals for their performance. In this sense, it can be noted that many of the time averages are in line with the NIC. Besides, some activities show lower averages than the NIC, probably because the nurses had experience in these specific clinical settings [3].

The scarcity of evidence found highlights the need to carry out research with greater methodological quality, such as prospective and retrospective observational or intervention studies with experimental methodology, and to broaden the samples to include new regions, contexts and settings in which NIC terminology can be applied. Similarly, extensive databases are required that contain information on these prevalences to establish new means of comparing the application of the interventions. It is essential to identify and map the activities carried out by nurses for the creation of lists [6,9] that provide the basis for procedural guidelines but also to understand the nature and typology of the interventions and their influence on the time required to perform them. The aim is to develop instruments that can be used to study nurses’ workloads [6,8,9]. Among the aspects that could affect these workloads is the highly specialised nature of care provided in units with clinical specificity, which appears to facilitate reductions in the time needed to successfully complete most of the interventions. However, the results reported indicate that the NIC averages could be lower than the times required by nurses to perform the interventions in up to 15% of cases. The idiosyncrasies of nursing care make it inevitable that the daily activity is disrupted by unscheduled interventions [3], as well as obstacles and constant interruptions to the nurses’ work, which affects up to 163.9% of the amount of time required to complete the interventions [15]. As such, the care routine imposed by work dynamics in the distinct care models can increase the activities assigned to nurses [16], which can affect the quality of care and health outcomes in the population.

The results are insufficient to consider that the averages indicated in the NIC should be adjusted to the clinical reality. Although these are generalised time averages, they only provide a basis to capture a valid measurement of nurses’ workloads.

The only experience that has been analysed to examine the intensity of its influence on nurses’ workloads corresponds to transfers of patients between facilities [5]. Although the results were lower than the averages established in the NIC, the authors considered that, during the transfer of the patients, the nurses did not record other activities involved in continuity of care or the documentation required for the administrative process of the patients’ transfers, which would increase workloads by up to 29%. 

On a further point, De Groot et al. [17] showed that nurses who work in the community setting are as likely to experience increases in their workloads when they carry out clinical documentation activities (mean = 8 h/week; SD = 6.0; median = 6.0) and when they perform organisational documentation activities (mean = 3.6 h/week; SD = 4.0; median = 2.0). However, no statistically significant correlation was observed between the time invested in clinical documentation and the perception of the increase in workloads (Spearman’s Rho 0.135; *p* = 0.058); while a statistically significant moderate relationship was seen between the time dedicated to organisational documentation and the perception of increased workloads (Spearman’s Rho 0.375; *p* < 0.000). This study concludes that the nurses who dedicate more time to organisational documentation have a greater propensity to perceive heavy workloads.

It should be pointed out that interest in studying these workloads with the use of the NIC was found in only a few nursing settings, such as oncology or paediatrics, and in a few clinical processes such as hospitalisation and surgical processes. Other limitations that affect the results and conclusions arise from inadequate mapping and coding of some of the interventions reported in studies, such as that carried out by Somensi et al. [4], which do not ensure that the averages indicated in these cases correspond closely to those in the NIC.

Although the eligibility criteria with JBI’s critical appraisal tools have been satisfied, study limitations are related to the methodological characteristics of the articles included; moreover, the heterogeneity and the absence of some statistical values in a few studies limit the validity of the results, which provide low levels of evidence.

## 5. Conclusions

The evidence found through the use of NIC terminology to determine nurses’ workloads is not conclusive. The NIC time averages are an adequate tool for understanding the impact of nurses’ workload on people’s health care. Yet the number of studies needs to be increased to provide more scientific evidence, along with improvements in methodological quality and rigour. Nurses must implement the quantity and quality of the recording in standardised NIC terminology throughout health records and in all clinical settings to advance the study of its relationship to the measurement of nurses’ workload. This could substantially contribute to improvements in staffing and quality of patient care.

## Figures and Tables

**Figure 1 healthcare-10-01141-f001:**
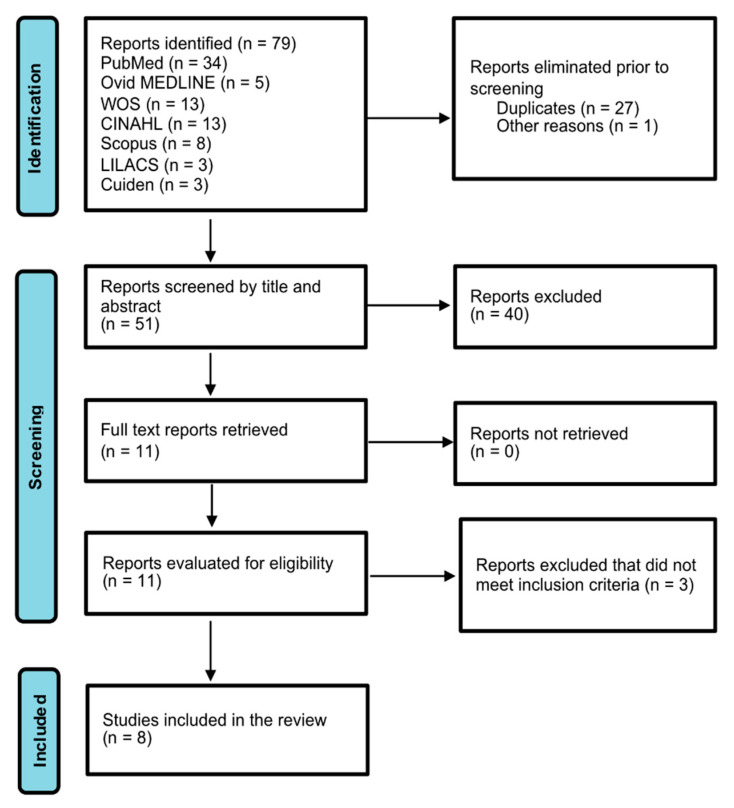
Flow diagram.

**Table 1 healthcare-10-01141-t001:** Author, Year, Journal, Country, Methodology and Study Aims.

Author (Year)	Journal	Setting (Country)	Methodology	Aim
Cordova et al. (2010) [3]	Journal of Nursing Care Quality	Orthopedic surgery (U.S.)	Qualitative	Determine the utility of NIC ^1^ terminology in classifying nursing care interventions as a measure of nursing workloads
Souza, Jericó & Perroca (2013) [9]	Revista Latino Americana de Enfermagem	Oncology (Brazil)	Descriptive	Identify the interventions and activities carried out in a chemotherapy centre, using standardised language, and validate its contents
Martin & Gaidzinski (2014) [6]	Einstein	Pediatric oncology (Brazil)	Descriptive	Create and validate an instrument to measure the time dedicated by nurses to interventions and activities in a Pediatric Hematology and Oncology Service Outpatient Centre
Assis et al. (2015) [7]	Revista da Escola de Enfermagem da USP	Pediatrics (Brazil)	Descriptive	Identify and validate the interventions and activities carried out by a nursing team in a pediatric unit in Brazil
Possari et al. (2015) [10]	Revista Latino Americana de Enfermagem	Oncology surgery (Brazil)	Descriptive	Analyse the distribution of nursing professionals’ workloads related to interventions and activities during the trans-operative period in a surgical centre specializing in oncology based on the NIC
Somensi et al. (2018) [4]	Revista Brasileira de Enfermagem	Hospital ward (Brazil)	Observational descriptive	Measure the workloads of nurses working on Hospital Wards, as recommended by the NIC, comparing observational and online methods to propose supervision strategies for professionals and academics
Sun, Li & Shen (2021) [8]	Studies in health technology and informatics	Pediatric oncology (China)	Descriptive	Identify and analyse the workloads of nursing professionals, according to the NIC, in a pediatric oncology centre
Trovó, Cucolo & Perroca (2021) [5]	Revista da Escola de Enfermagem da USP	Hospital ward (Brazil)	Observational descriptive	Measure the average time taken by nurses to transfer patients; compare the activities observed during this intervention with those described in the NIC and explore the intensity of their influence on workloads

^1^ Nursing Interventions Classification.

**Table 2 healthcare-10-01141-t002:** Number of activities and interventions identified and validated.

Author (Year)	Activities Identified	Activities Validated	NIC ^1^ Identified	NIC Validated
Cordova et al. (2010) [3]	-	-	224	42
Souza, Jericó & Perroca (2013) [9]	48	48	35	35
Martin & Gaidzinski (2014) [6]	-	-	32	25
Assis et al. (2015) [7]	275	205	63	53
Possari et al. (2015) [10]	266	266	49	49
Somensi et al. (2018) [4]	-	-	30	-
Sun, Li & Shen (2021) [8]	13,021	-	89	-
Trovó, Cucolo & Perroca (2021) [5]	-	-	-	-

^1^ Nursing Interventions Classification.

**Table 3 healthcare-10-01141-t003:** Type of activities.

Author (Year)	Scheduled	Unscheduled	Direct (Included in NIC ^1^)	Indirect (Included in NIC)	Associated	Personal
Cordova et al. (2010) [3]	25 NIC (59.53%)	17 NIC (40.47%)	-	-	-	-
Assis et al. (2015) [7]	-	-	238 (86.54%)	24 (8.73%)	13 (4.73%)	
Possari et al. (2015) [10]	-	-	380 (42.79%)	373 (42%)	71 (8%)	64 (7.21%)
Somensi et al. (2018) [4]	-	-	16 NIC (face-to-face: 402 activities; telematic: 334 activities)	14 NIC (face-to-face: 1499 activities; telematic: 1175 activities)	-	-
Sun, Li & Shen (2021) [8]	-	-	63 NIC (35.84% activities)	26 NIC (43.66% activities)	-	-

^1^ Nursing Interventions Classification.

**Table 4 healthcare-10-01141-t004:** Prevalence of interventions identified in the studies in numerical or alphabetical order.

Interventions	Type of Activity	Possari et al. (2015) [10] *n* (%)	Somensi et al. (2018) [4] (In-Person Observation) *n* (%)	Somensi et al. (2018) [4] (Telematic Observation) *n* (%)
0580 Bladder catheter	Direct	18 (2.03)	4 (1.0)	3 (0.90)
0582 Intermittent bladder catheterization	Direct	-	1 (0.25)	11 (3.29)
0842 Change of position: intraoperative	Direct	14 (1.58)	-	-
1080 Nasogastric tube	Direct	-	5 (1.24)	3 (0.90)
1630 Emotional support	Direct	2 (0.10)	-	-
1806 Help with self-care: transfer	Direct	17 (1.91)	-	-
1872 Thoracic drain care	Direct	-	28 (6.97)	16 (4.79)
2000 Electrolyte management	Direct	9 (1.01)	-	-
2300 Administration of medication	Direct	-	10 (2.49)	15 (4.49)
2870 Post-anesthesia care	Direct	6 (0.69)	-	-
2900 Surgical assistance	Direct	49 (5.52)	-	-
2920 Surgical precautions	Direct	41 (4.62)	-	-
3160 Airway suction	Direct	-	2 (0.50)	2 (0.60)
3660 Wound care	Direct	1 (0.11)	-	-
3902 Temperature regulation: intraoperative	Direct	12 (1.35)	-	-
4030 Administration of blood products	Direct	2 (0.22)	-	-
4054 Central venous access device management	Direct	-	19 (4.73)	13 (3.89)
4130 Liquid monitoring	Direct	1 (0.11)	2 (0.50)	-
4232 Phlebotomy: arterial blood extraction	Direct	-	15 (3.73)	8 (2.40)
4238 Phlebotomy: venous blood extraction	Direct	-	6 (1.49)	4 (1.20)
4820 Orientation in the health system	Direct	-	230 (57.21)	150 (44.91)
5340 Presence	Direct	31 (3.49)	-	-
5460 Contact	Indirect	-	29 (1.93)	40 (3.40)
6200 Emergency care	Direct	-	27 (6.72)	26 (7.78)
6320 Reanimation	Direct	-	-	2 (0.60)
6482 Environmental management: comfort	Direct	4 (0.45)	-	-
6486 Environmental management: safety	Direct	2 (0.22)	-	-
6545 Infection control: intraoperative	Direct	19 (2.14)	-	-
6650 Vigilance	Direct	76 (8.56)	-	-
6680 Monitoring vital signs	Direct	1 (0.11)	-	-
7140 Family support	Direct	28 (3.15)	-	-
7640 Development of clinical pathways	Indirect	1 (0.11)	-	-
7650 Delegation	Indirect	76 (8.56)	-	-
7680 Help in exploration	Direct	-	29 (7.21)	53 (15.87)
7710 Cooperation with the doctor	Indirect	11 (1.24)	-	-
7760 Product assessment	Indirect	2 (0.22)	-	-
7820 Sample management	Indirect	3 (0.35)	-	-
7830 Supervision of personnel	Indirect	-	78 (5.20)	119 (10.13)
7840 Supply change management	Indirect	32 (3.60)	-	-
7850 Personnel development	Indirect	18 (2.03)	27 (1.80)	27 (2.30)
7880 Management of technology	Indirect	1 (0.11)	-	-
7892 Transport: within the facility	Direct	49 (5.52)	-	-
7920 Documentation	Indirect	166 (18.69)	-	-
7960 Exchange of health care information	Indirect	-	370 (24.68)	120 (10.21)
8140 Transfer of patient care	Indirect	63 (7.09)	-	-
Billing review *	Indirect	-	10 (0.67)	9 (0.77)
Contact with medical staff *	Indirect	-	185 (12.34)	45 (3.83)
Control of psychotropic medication stock *	Indirect	-	18 (1.20)	32 (2.72)
Filling in protocols *	Indirect	-	27 (1.80)	13 (1.11)
Medical record review *	Indirect	-	114 (7.61)	35 (2.98)
Peripheral vein catheterisation *	Direct	-	16 (3.98)	21 (6.29)
Prescription registration *	Indirect	-	265 (17.68)	384 (32.68)
Recording medical history *	Indirect	-	25 (1.67)	22 (1.87)
Recording patient course *	Indirect	-	296 (19.75)	302 (25.70)
Requesting material *	Indirect	-	29 (1.93)	14 (1.19)
Withdrawal of central vascular catheter *	Direct	-	8 (1.99)	7 (2.10)
Worksheet preparation *	Indirect	-	26 (1.73)	13 (1.11)

* Interventions not identified with a NIC code.

**Table 5 healthcare-10-01141-t005:** Average time observed in the studies and NIC averages in minutes in numerical or alphabetical order.

Interventions	Cordova et al. (2010) [3] *n* (CI ^1^)	Somensi et al. (2018) [4] (In-Person Observation) *n* (CI)	Somensi et al. (2018) [4] (Telematic Observation) *n* (CI)	Average NIC ^2^
0140 Encourage body mechanics	15.08 (5–30)	-	-	16–30
0450 Management of constipation/fecal impactation	15.90 (15–45)	-	-	16–30
0580 Bladder catheter	17.61 (3.5–30)	7.19 (7.19–7.19)	16.92 (5.8–28.06)	<15
0582 Intermittent bladder catheter	-	15.45 (15.5–15.5)	10.77 (6.3–15.27)	<15
0740 Care of bedridden patient	11.96 (3–30)	-	-	16–30
0910 Inmobilisation 0940 Traction/immobilisation care	15.67 (2.5–60)	-	-	<15
1080 Nasogastric tube	-	23.59 (13.8–33.4)	5.01 (0–12.26)	<15
1450 Management of nausea 1570 Management of vomiting	14.25 (2–45)	-	-	16–30
1806 Help with self-care: transfer	18.98 (3.5–45)	-	-	<15
1872 Thoracic drain care	-	4.97 (3.51–6.43)	8.88 (4.8–13.02)	<15
2080 Management of liquids/electrolytes	19.66 (1.5–60)	-	-	<15
2300 Administration of medication	27.11 (2–120)	6.11 (2.78–9.44)	8.44 (5.2–11.77)	<15
2620 Neurological monitoring	11 (1–30)	-	-	16–30
2690 Seizure precautions	14.62 (2–30)	-	-	16–30
2930 Surgical preparation	29.35 (7.5–75)	-	-	46–60
3160 Airway suction	-	0.38 (0.38–0.38)	7 (0–52.74)	<15
3590 Skin exploration	11.89 (2–30)	-	-	16–30
3660 Wound care	22.65 (4–60)	-	-	31–45
4020 Bleeding reduction	17.47 (1.5–60)	-	-	46–60
4030 Administration of blood products	31.08 (5–60)	-	-	>60
4054 Central venous access device management	-	5.22 (1.17–9.28)	19.21 (9.8–28.6)	31–45
4110 Precautions in embolism	13.47 (2–30)	-	-	16–30
4130 Monitoring liquids	-	5.05 (5.05–5.05)	-	16–30
4210 Invasive hemodynamic monitoring	22.71 (5–120)	-	-	46–60
4232 Phlebotomy: arterial blood extraction	-	8.87 (2.72–15.03)	8.91 (4.8–13.02)	<15
4238 Phlebotomy: venous blood extraction	-	7.36 (6.19–8.53)	11.44 (0–24.58)	<15
4820 Orientation in the health system	-	3.16 (3.01–3.31)	4.44 (3.64–5.24)	16–30
5240 Counselling	22.04 (5–45)	-	-	46–60
5460 Contact	-	29.9 (11.9–47.91)	28.17 (24.5–31.9)	<15
5606 Education: individual	21.91 (4–60)	-	-	31–45
6200 Emergency care	40.52 (2–120)	18.1 (0–52.2)	9.09 (0–11.52)	16–30
6320 Reanimation	-	-	32.72 (0–233.7)	16–30
6460 Dementia management	35.50 (1.5–120)	-	-	>60
6486 Environmental management: safety	-	16.74 (11.86)	16.74 (11.86)	31–45
6540 Infection control	14.42 (1.5–30)	-	-	31–45
7310 Nursing care on admission	38.31 (5–75)	-	-	16–30
7370 Planning for discharge	31.70 (4–60)	-	-	46–60
7680 Assistance in exploration	14.85 (2–45)	6.78 (5.7–7.86)	4.27 (3.9–4.62)	16–30
7830 Supervision of personnel	-	2.07 (1.87–2.28)	7.21 (2.99–11.4)	>60
7850 Personnel development	-	14.5 (0–58.52)	1.73 (1.6–1.87)	>60
7910 Consultation	24.96 (4–62)	-	-	46–60
7960 Health care information exchange	22.34 (3–60)	-	-	<15
Billing review *	-	12.41 (1.6–23.22)	13.2 (6.86–19.5)	46–60
Contact areas of support *	-	1.36 (1.34–1.39)	3.38 (2.86–3.89)	<15
Contact with medical personnel *	-	1.49 (1.28–1.69)	3 (2.38–3.64)	16–30
Control of psychotropic medication *	-	8.11 (0–18.78)	12.06 (7.67–16.5)	16–30
Filling-in protocols *	-	5.41 (3.18–7.65)	7.59 (4.06–11.1)	46–60
Medical record review *	-	5.51 (5.39–5.63)	31.19 (20.3–42.1)	46–60
Peripheral vein catheterization *	-	8.71 (0–57.9)	11.72 (7.9–15.56)	31–45
Prescription registration *	-	3.18 (2.97–3.38)	3.03 (2.8–3.27)	16–30
Recording medical history *	-	9.48 (9.17–9.78)	10.19 (7.3–13.1)	16–30
Recording patient course *	-	2.32 (2.06–2.59)	3.43 (3.16–3.7)	16–30
Requesting material *	-	5.39 (0–11.52)	11.88 (9.41–14.3)	16–30
Withdrawal of central vascular catheter *	-	9.06 (5.89–12.2)	9.13 (6.9–11.39)	31–45
Worksheet preparation *	-	18.81 (0–219.8)	36.97 (25.9–48)	46–60
1410 Pain management: acute **	13.39 (2–45)	-	-	31–45
0960 Transport ***	27.97 (4–75)	-	-	<15

^1^ Confidence interval; ^2^ Nursing Interventions Classification; * Interventions not identified with a NIC code; ** Label modified in the NIC 7th edition; *** Label not available in the NIC 7th edition [1].

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
