# Peer review of "Use of the Nursing Interventions Classification and Nurses’ Workloads: A Scoping Review"

_healthcare, 2022, doi:10.3390/healthcare10061141_

Round 1
Reviewer 1 Report
Dear Authors,
I read your paper with interest because it touches on the issue of nursing workload. With the increasing shortage of nurses in the world, this is a very important topic.
I have a few comments about the research and the paper:
1 - I do not understand why in table 4 you have collated the results of only two groups of researchers. Why did you use the criteria of the highest number of interventions and the lowest number of interventions to guide your selection?
Data records in the table are different, e.g: Possari et al. - 18 (2.03) Somensi et al. - P 1 : 4 (1.0); T 2 : 3 (0.90)
2 - the same remarks apply to Table 5
3 - the discussion is quite general
4 - what are the limitations of the study?
5 - what about the conclusions? They are too general. Why investigate further? What is it for? I think this needs to be made clearer because it is very important
Author Response
Additional corrections:
Firstly, we have detected a typo in the surname of one of the authors that is important to correct. The right name of the author with the typo should be: Domingo-Ángel Fernández-Gutiérrez.
We have corrected some typographical errors in the text, which have been highlighted in red.
Reviewer 1
I read your paper with interest because it touches on the issue of nursing workload. With the increasing shortage of nurses in the world, this is a very important topic.
The authors would like to acknowledge the comments made on the article.
I have a few comments about the research and the paper:
1 - I do not understand why in table 4 you have collated the results of only two groups of researchers. Why did you use the criteria of the highest number of interventions and the lowest number of interventions to guide your selection?
A sentence has been included to clarify that data on the prevalence of interventions have been reported by only two articles. Interventions have been sorted in number order and alphabetically as appropriate.
Data records in the table are different, e.g: Possari et al. - 18 (2.03) Somensi et al. - P 1 : 4 (1.0); T 2 : 3 (0.90)
We are grateful for comments on the content that allowed us to improve the table. As Somensi et al.'s study reported data on the prevalence of interventions in two different observations (in-person observation and telematic observation), to improve comprehension we have separated the data into two columns.
2 - the same remarks apply to Table 5
We have made the same modifications as in the previous table to improve the comprehensibility of the table.
3 - the discussion is quite general
We have added comments to the results of the review in order to be more specific in the discussion of the results.
4 - what are the limitations of the study?
We have included the limitations of the study at the end of the discussion.
5 - what about the conclusions? They are too general. Why investigate further? What is it for? I think this needs to be made clearer because it is very important
We have added additional information in the conclusions.

Reviewer 2 Report
You have identified an important area of study. The fact that only 8 articles relevant to the question is an indication of the need for this work as a baseline foundation for continued study of the question posed. Excellent work!
Author Response
Additional corrections:
Firstly, we have detected a typo in the surname of one of the authors that is important to correct. The right name of the author with the typo should be: Domingo-Ángel Fernández-Gutiérrez.
We have corrected some typographical errors in the text, which have been highlighted in red.
Reviewer 2
You have identified an important area of study. The fact that only 8 articles relevant to the question is an indication of the need for this work as a baseline foundation for continued study of the question posed. Excellent work!
The authors would like to acknowledge the comments made on the article.

Reviewer 3 Report
Dear Authors,
This is an interesting and well-written paper referring to NIC.
I hereby enclose some suggestions that you might include for the improvement of the paper.
Introduction is well written. Please make sure that the rationale of the study is clearly highlighted and previously published work in this context is reported. Maybe some additional references would help to enrich the background information and the bibliographical sources included.
In the methods section it would be also helpful to write some more information on scoping review and justify why doing a scoping review is most appropriate for this work. You may further explain why the objectives of your work and the review question as posed, led you to choose the scoping review approach.
Please, check and make sure that the term scoping review is used throughout the paper.
In the discussion part, please make sure that you discuss in an explicit way the process used for examining the studies included in your work, in terms of methodological rigour and validity, so your answer to your review question is clearly supported and presented.
Please check the paragraph in lines 207 – 211, as the meaning of it is not very clear.
In the Conclusion you may also refer to potential implications for practice and research
Kind regards
Author Response
Additional corrections:
Firstly, we have detected a typo in the surname of one of the authors that is important to correct. The right name of the author with the typo should be: Domingo-Ángel Fernández-Gutiérrez.
We have corrected some typographical errors in the text, which have been highlighted in red.
Reviewer 3
Dear Authors,
This is an interesting and well-written paper referring to NIC.
The authors would like to acknowledge the comments made on the article.
I hereby enclose some suggestions that you might include for the improvement of the paper.
Introduction is well written. Please make sure that the rationale of the study is clearly highlighted and previously published work in this context is reported. Maybe some additional references would help to enrich the background information and the bibliographical sources included.
We believe that the rationale for our work is to explore the available evidence on the use of NIC for the identification and measurement of nurses' workload since the most representative previous reference works on this topic are scarce and inconclusive. We are unable to add further references in the introduction as to the best of our knowledge there is no one; to improve this, we have included in the introduction some of the bibliographical references to the studies included in the review themselves. The rationale for the scoping review has been more clearly stated.
In the methods section it would be also helpful to write some more information on scoping review and justify why doing a scoping review is most appropriate for this work. You may further explain why the objectives of your work and the review question as posed, led you to choose the scoping review approach.
We have explained these aspects in the methodology in more detail and added two bibliographical references to support the explanation of the scoping review methodology.
Please, check and make sure that the term scoping review is used throughout the paper.
The term "Scoping" has been included in all relevant places to refer to the methodology of the review.
In the discussion part, please make sure that you discuss in an explicit way the process used for examining the studies included in your work, in terms of methodological rigour and validity, so your answer to your review question is clearly supported and presented.
We have added discussion of the results of the studies included in the review. By way of limitations of the study, we have added discussion on the process of critical reading about the characteristics of the studies that can be included in a scoping review.
Please check the paragraph in lines 207 – 211, as the meaning of it is not very clear.
There must be an error as lines 207 to 211 correspond to Table 5 and not to a paragraph. Additionally, we have improved the content of tables 4 and 5 to make them more comprehensive.
In the Conclusion you may also refer to potential implications for practice and research
We have supplemented the conclusions of the study by adding implications for clinical practice.
Kind regards
